# The rising moon promotes mate finding in moths

Mona Storms [1,4], Aryan Jakhar [2], Oliver Mitesser [1], Andreas Jechow [3], Franz Hölker [3], Tobias Degen [1], Thomas Hovestadt [1] & Jacqueline Degen [1,4 ✉]

To counteract insect decline, it is essential to understand the underlying causes, especially for key pollinators such as nocturnal moths whose ability to orientate can easily be influenced by ambient light conditions. These comprise natural light sources as well as artificial light, but their specific relevance for moth orientation is still unknown. We investigated the influence of moonlight on the reproductive behavior of privet hawkmoths (*Sphinx ligustri*) at a relatively dark site where the Milky Way was visible while the horizon was illuminated by distant light sources and skyglow. We show that male moths use the moon for orientation and reach females significantly faster with increasing moon elevation. Furthermore, the choice of flight direction depended on the cardinal position of the moon but not on the illumination of the horizon caused by artificial light, indicating that the moon plays a key role in the orientation of male moths.

[1] Biocenter of the University of Würzburg, Würzburg, Germany. [2] School of Biology, Indian Institute of Science Education and Research Thiruvananthapuram, Thiruvananthapuram, India. [3] Leibniz Institute of Freshwater Ecology and Inland Fisheries, Berlin, Germany. [4] These authors contributed equally: Mona Storms, Jacqueline Degen. ✉email: jacqueline.degen@uni-wuerzburg.de

The accelerating decline of insects has become a major topic in nature conservation and research over the last years[1,2]. Principal stressors include land use and climate change, agriculture, introduced species, nitrification, and pollution[3]. Along with the general insect decline, a significant decrease in abundance and altered distribution of moths has been observed across Europe[4]. Moths play a crucial role as nocturnal pollinators and are important components of almost all terrestrial food webs[4–6]. Consequently, there is an urgent need to understand the reasons for their decline. Recently, artificial light at night (ALAN), or "light pollution", has been suggested as a possible driver for insect decline in general[7,8] and the decline of nocturnal moths in particular (see discussion in[4,9,10]). Nocturnal insects have evolved under natural nocturnal light conditions and can therefore utilize dim light including starlight for orientation[11–14]. For example, dung beetles can use the Milky Way as an orientation cue[15] and also sense polarization patterns from moonlight[16,17].

ALAN has become a global threat, growing in intensity and affecting increasingly larger areas[18,19]. It alters natural light regimes with potential long-term effects on nocturnal insects[10,20]. Recently, it was shown that dung beetle behavior is affected by ALAN[21] and the impact of ALAN on important ecosystem services such as nocturnal pollination was documented[22]. However, the underlying mechanisms and cues of the nocturnal orientation of moths are still poorly understood. In particular, the nocturnal orientation in the context of mate finding remains largely unknown but is of utterly importance as the survival and mating success of moths might decrease by ALAN-mediated degradation of such orientation cues[23]. The most easily perceived celestial body during the night is the moon. However, due to its variable, temporally limited visibility the moon is more difficult to utilize as compass compared to the sun[12]. Nevertheless, moonlight can potentially serve nocturnal insects for orientation[11].

In this study, we combined behavioral experiments performed with free-flying male moths (*Sphinx ligustri*) with a detailed quantification of the nocturnal light environment using an all-sky camera. This allowed us to study natural light sources like the moon and the stars as well as "skyglow"—a type of indirect light pollution that originates from light radiated upwards that is then scattered back within the atmosphere[19]. We find that the visibility of the moon above the horizon improves the ability of male moths to find females and that they succeed faster as the moon rises. Although the moon increases the brightness of the entire environment, the cardinal position of the moon significantly influences the flight direction of males, as they choose to fly more frequently towards (caged) females located in the same hemisphere as the moon. Since bright areas at the horizon illuminated by distant light sources or skyglow do not trigger a comparable behavior, the moon as a natural light source apparently plays a key role in the orientation of male moths.

## Results and discussion

### The moon increases mate finding in moths.

To investigate the impact of natural and artificial light sources on mate finding, we analyzed flight behavior in male moths, which were reliably attracted by caged virgin females (see Materials and Methods for details). Since we used these females specifically to exploit their attraction effect, we refer to them as 'traps' in the following. To establish a choice scenario (see below), males were released equidistantly from the traps, which were located north and south of the core release site in central Germany. Besides the stars, the moon creates the natural light environment that moths might use for visual orientation. We therefore first tested if the moon affects mate finding. We found that the percentage of males arriving within the experimental time (8 min from release, 58.6% of

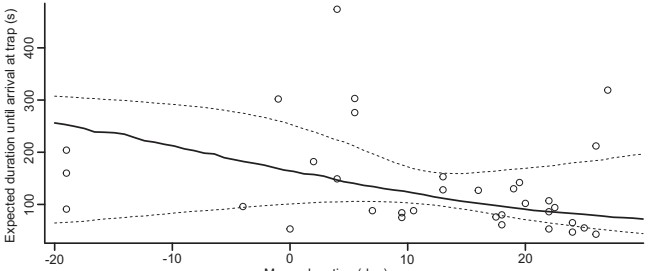

**Fig. 1 Expected flight duration of a moth.** Flight duration (black line) was calculated as the median flight duration predicted by the Cox PH model ($p = 0.014$, $n = 34$) for arrivals within 8 minutes after release and averaged over all individuals. Circles represent the actual measured values. Dashed lines indicate the confidence interval of the predicted duration at $\alpha = 5\%$ level estimated by bootstrapping (5000 replicates).

flights) at a trap increased significantly with the appearance of the moon (logistic regression: $z = -2.06$, $p = 0.04$, $n = 58$) and did not depend on the presence of clouds in front of the moon ($z = -0.83$, $p = 0.406$, $n = 58$). A few males reached the females later during the experimental night (13.8% of flights) and were released again on the next day. Some males never reached a trap and could therefore not be tested again in the next days (27.6% of flights). Furthermore, the time that successful males needed to reach a trap was significantly influenced by the height of the moon above or below the horizon (Fig.1; Cox PH survival model, $z = 2.46$, $p = 0.014$, $n = 34$): the higher the moon was above the horizon, the faster males were able to locate and reach the females. The presence of clouds in front of the moon did not play a significant role in this context either ($z = -0.65$, $p = 0.519$, $n = 34$), leading to the conclusion that the moon was equally well perceived if covered partly by clouds and used for effective orientation towards the females. Although the lunar phase changed during the period of the experiment from full moon to new moon, flight duration was not significantly affected by the percentage of the lit moon disk ($z = 0.44$, $p = 0.66$, $n = 34$). Thus, the properties of the moon that affected the flight duration of males were independent of the lunar phase.

It is important to emphasize that the results were not significantly affected by traits on the individual level like body size or origin of the animal (see Supplementary Results and Discussion for details). Furthermore, a possible learning effect of animals that were released more than once was not detectable since flight duration did not decrease depending on 'experience' but only with the elevation of the moon (Fig. S1). Thus, the moon as an easily perceivable orientation cue increased mate finding in general but also depended on its elevation. Despite two exceptions of long flight durations at moon elevations > 20° that go back to the same animal probably for individual reasons (Fig. S1), the variance in flight duration was highest at low moon elevations (Fig. 1). This relatively high variance at low moon elevations emphasizes the question if artificial lights affected mate finding, particularly whenever the moon as a natural light cue was not yet prominent.

### Linking flight behavior to the light environment.

We used a calibrated digital all-sky camera to track changes in the natural and artificial components of the night sky brightness[24] (Fig. 2 a–c). A similar camera system was recently used to study dung beetle behavior[21]. Although the impact of light pollution on the site was not strong, the night sky was also not completely pristine. Luminance ($L_{Vv}$) values were about 0.34 mcd/m² at zenith and 1.6 mcd/m² near the horizon under clear sky conditions when the moon was not visible. A natural (unpolluted) sky brightness is 0.25 mcd/

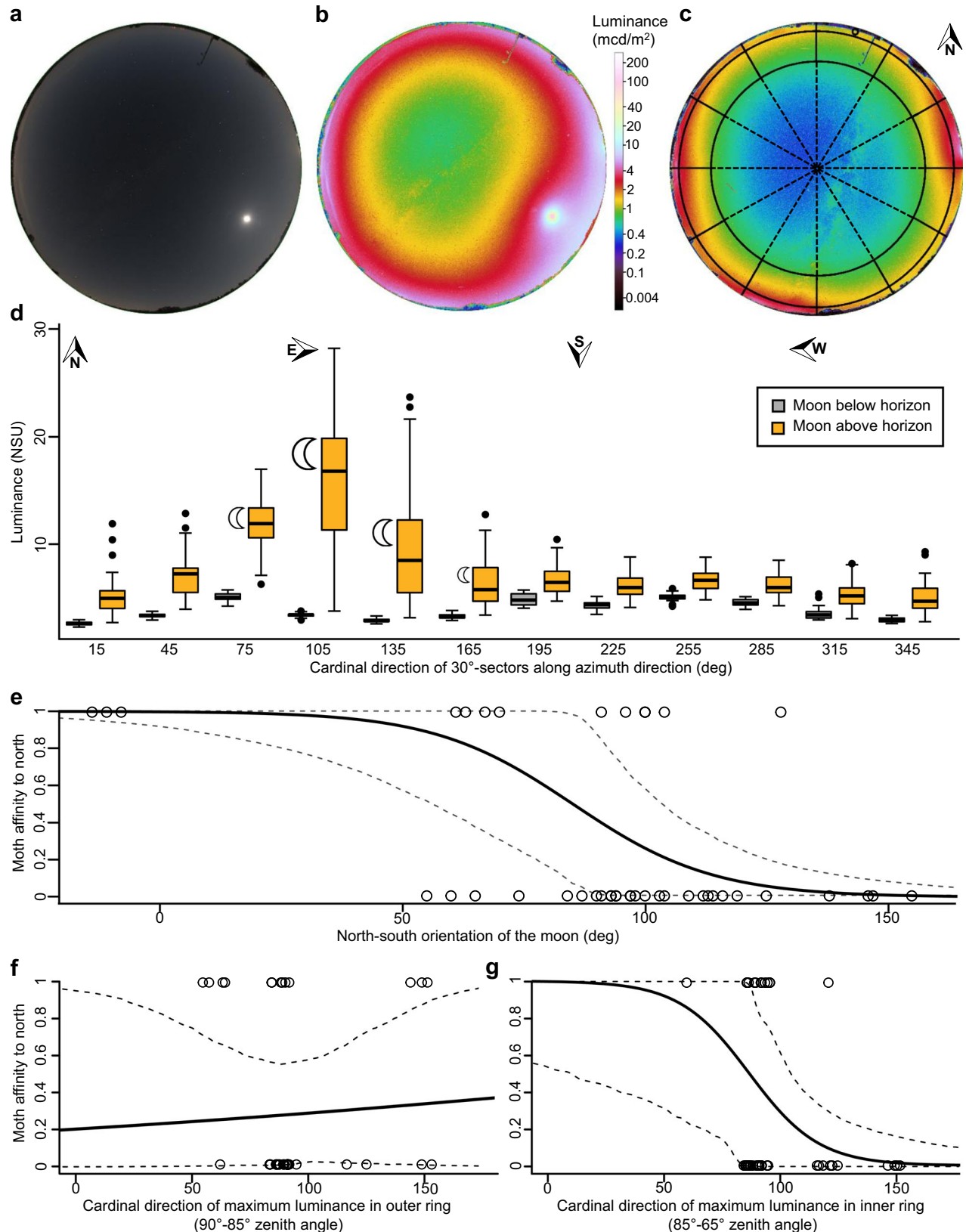

m² at zenith and can be used as the reference value "Natural Sky Unit" (NSU) for easy comparison (see also Materials and Methods). The analysis of specific sky sectors revealed that the moon was the strongest factor determining the ambient brightness, brightening every sector of the sky as soon as it appeared above the horizon (Fig. 2d). During observation times, the course of the moon mainly progressed through the eastern part of the sky, affecting particularly the $L_{vV}$ values in the corresponding sectors (Fig. 2d). Furthermore, light conditions never corresponded to a non-light polluted sky, as NSU values were always greater than one. Most sectors in the south, west and north (sectors seven to 12 and one) were hardly subjected to fluctuations. Nevertheless, it is

**Fig. 2 Quantification of the light environment with all-sky imagery and its impact on flight behavior of moths. a** Raw RGB all-sky image with clear sky and a visible moon 26° above the horizon at 119° azimuth angle, South-east (24 July 2019, 03:23). **b** Same image as in **a** with processed luminance values. **c** Processed all-sky image in luminance with clear sky, a visible milky way (green patches in a 'ribbon-shape' across the (blue) night sky), skyglow near the horizon, and a non-visible moon 0° above the horizon at 87° azimuth angle, East (24 July 2019, 0:25). The colors of the processed image correspond to the legend in **b**. The black lines mark the sky segments used to quantify the light environment. The outer ring covers 5° above the horizon (85°−90° zenith angle), the inner ring 20° above the outer ring (65°−85° zenith angle). Furthermore, the sky was divided into 12 sectors of 30° width along the azimuth direction (extension by dashed line), starting with the sector marked with the small circle (counting clockwise). **d** Luminance in natural sky units (NSU) for each full sector of 30°. The moon icons indicate sectors in which the moon was visible, regardless of its phase. The size of each symbol encodes the rank of the frequency ($n = 33$). **e** Trap choice of arrived males depending on the position of the moon at the moment of release on the north-south axis (north = 0°). The y-axis displays choice of the southern trap at 0.0 and of the northern trap at 1.0. $p = 0.022$, $n = 42$. **f** Male moth affinity to northern trap in response to the direction of maximum luminance measured in the outer ring of 5°. Each circle indicates an observed arrival, $p = 0.753$, $n = 41$. **g** Male moth affinity to northern trap as in **f** but with luminance measured in the inner ring of 20°, $p = 0.065$, $n = 41$. **e–g** The line represents the prediction of the logistic model, providing a probability value for arriving at the northern trap (north prone = 1; south prone = 0). Dashed lines indicate the confidence interval of the prediction at α = 5% level estimated by bootstrapping (5000 replicates).

recognizable that the moon made a decisive contribution to the light environment in all directions since images with the moon above the horizon were always brighter than those with the moon below the horizon (Fig. 2d).

Due to the design of the experiment with one trap located in the north and the other in the south of a central release site, we were able to investigate the choice behavior of males, especially in respect of the possible influence of the cardinal position of the moon as it was almost exclusively visible in the southern hemisphere of the sky (Fig. 2d). Although the moon continued to move south during the night, the moon's cardinal position never overlapped with the exact direction of the southern trap. The only parameter that had a significant effect on choice behavior was indeed the cardinal position of the moon (Fig. 2e, logistic regression, $z = -2.3$, $p = 0.022$, $n = 42$). The more southern the moon's position was, the more likely males flew to the southern trap. However, while some clouds in front of the moon had no significant effect on choice behavior ($z = 0$, $p = 1$, $n = 42$), moon above the horizon showed a tendency to affect males ($z = -1.82$, $p = 0.069$, $n = 42$). The results indicate that despite the general increase of ambient brightness by the moon, it is its position that mainly influenced the flight direction of males. Thus, moths preferred a flight direction with the prominent compass cue ahead to steer their flight towards the females but it is important to emphasize that moon and trap had an angular difference of at least 23° (80.8° to the moon's mean cardinal direction). Therefore, males that chose to fly towards the southern trap did not fly directly towards the direction of the moon.

As the moon represents a natural distant light source, we tested whether distant artificial light sources or skyglow might elicit a comparable effect on the behavior of male moths and if such light sources might mask the moon. To evaluate the light environment with regards to these aspects, we defined sky segments of particular interest that occurred due to the location of the experimental field (Fig. 2c). For each arrival at a trap, the brightest sector of the environment was determined and placed on a north-south axis of maximum 180 degrees (Fig. 2f, g). If we look at the brightest sector of the environment and distinguish between the area close to the horizon, i.e. "outer ring" (Fig. 2f) and the one above, i.e. "inner ring" (Fig. 2g), we can observe differences in trap choice. The line indicates the logistic regression model and provides the probability of arriving at the northern trap. For the $L_v$ in the area close to the horizon no effect of maximum $L_v$ on trap choice could be found (logistic regression, $z = 0.31$, $p = 0.753$, $n = 41$). For the segment further above the horizon the probability of flying to the southern trap increased with maximum $L_v$ but the results are marginally not significant ($z = -1.85$, $p = 0.065$, $n = 41$). Our results for trap selection indicate that distant artificial lights of the surroundings

did not attract males and support the hypothesis that the moon, once it appears above the horizon and stands out from the general light (pollution) near the horizon (above five degrees), is used as an effective visual cue with moths rather flying towards than away from.

Digital cameras are suitable to measure the dynamics of night-time lighting conditions[25,26], and allow researchers to track changes in artificial lighting conditions and brightness of the sky simultaneously[27]. However, it is not straightforward to distinguish between ALAN and natural light sources like the moon with luminance images when the moon is close to the horizon and thus in the section of the sky where most light pollution occurred. Yet, once the moon rose higher than 5° and thus stood out distinctly from the light-polluted horizon, it could be clearly identified on the images (Fig. 2b). In this context, it is particularly remarkable that the speed at which the females were reached increased reliably only above a similar threshold (Fig. 1), with the only exceptions of two flights with long durations at a moon elevation greater than 20° (Fig. 1); both flights originated from the same individual (Fig. S1). Thus, the high variance of flight durations at low moon elevations (Fig. 1) supports our hypothesis that the moon, as an orientation cue, can be masked by artificial light for the animals as well. Yet, this hypothesis needs to be explicitly tested in future experiments. In general, the possible consequences of light pollution are still uncertain[28], especially because the amount of artificial light emitted during the night continues to increase exponentially worldwide[18]. But regardless of this, the moon is the decisive orientation cue as soon as it is visibly silhouetted against the horizon despite possible diffuse light pollution.

Another interesting next research project would be to investigate the relevance of polarized light, as this could provide an explanation for the occasional fast flights at times of low lunar elevations (cf. Figure 1). Furthermore, it might explain why flight duration was not significantly affected by clouds in front of the moon since the polarization pattern extends over the whole sky and is therefore not shielded completely by scattered clouds[29]. For dung beetles it has been already shown that they are capable of using the polarization signal for navigation[16,30,31] and it has been proposed that moths might be capable of utilizing the same signal[32]. At the same time, it has already been demonstrated that urban skyglow can diminish the lunar polarization signal[33], making a detailed investigation of the interplay between these two factors and the significance for moth orientation particularly exciting to understand underlying mechanisms.

Our results confirm that moths use the moon as an orientation cue, supporting the notion of Vickers & Baker[34] that pheromones alone are not sufficient for successful (and fast) orientation. Since flight duration decreased as a function of lunar elevation, we

conclude that the moon contributes to mating success, especially when it can be easily perceived. Since nocturnal landscapes around the world have been drastically restructured in terms of light intensity and light spectrum due to the rapid spread and increase of electrical lighting[18], a deeper understanding of orientation mechanisms even in the absence of the moon as an easily perceivable cue could provide a valuable contribution to counteract insect decline.

## Materials and Methods

**Experimental design**. The study was conducted from 19 July 2019 to 31 July 2019 on a meadow located east of Großseelheim and south of the river Ohm in the German State Hesse (50°49'17.5"N 8°52'15.4"E). The experiment was performed during the lunar phases from the full moon (19 July, 95.7% lit moon disk) to the new moon (31 July, 0.2% lit moon disk). A skyglow model based on satellite data suggests that the night sky brightness at zenith was relatively low (0.3 mcd/m²)[19], which is indicated by the visibility of the Milky Way (Fig. 2c) and confirmed by our own measurements from such all-sky imagery for moonless clear nights (0.34 mcd/m² ≈ 1.36 NSU). Skyglow is a type of indirect light pollution that originates from light radiated upwards that is then scattered back within the atmosphere[19]. The surrounding villages and towns utilize standard artificial light sources (streetlights, private lights etc.). Therefore, some direct light sources, as well as skyglow were visible at the (distant) horizon from our site, originating from the adjacent village Großseelheim (0.4 km), the nearby cities Kirchhain (4.2 km), Amöneburg (5.0 km), Marburg (7.8 km), Stadtallendorf (10.0 km), and Giessen (30.0 km). It is important to note that the setup was located far beyond the attraction radius, i.e. the distance around a lamp at which an animal is directly attracted towards the light source, of any artificial light in the surroundings, as these were much further away than 23 m[35].

Privet hawk moth males (*Sphinx ligustri* L., Lepidoptera, Sphingidae) were released individually and equidistantly (105 m) to cages with virgin females that served as pheromone traps. Each trap consisted of three virgin females kept individually in gauze cages attached to a wooden stick at a height of 1 m. To create a choice scenario, one trap was located in the North, the other one in the South of the release site. By selecting the north-south axis we took advantage of the circumstance that whenever the moon was above the horizon, it was almost exclusively visible in the southern hemisphere of the sky (85% of flights). The choice of the trap therefore also reflected whether males had chosen the hemisphere with the moon and thus indicated whether the cardinal position of the moon influenced flight behavior.

16 of the 23 males tested came from our own breeding, comprising the offspring of a mated female captured in the same study area one year before. These pupae raised were stored during winter in the refrigerator at a temperature of 5 °C and removed five weeks before the experiment. For hatching, the pupae were placed on the bottom of a cardboard box, as this allowed them to climb the walls and unfold their wings easily. Spatial separation was used to ensure that males and females were unable to mate. The hatching boxes were kept in a room without thermoregulation, so that humidity and temperature fluctuated between day and night according to the warm summer temperatures. The other seven males were attracted by the pheromones of females and caught in the field. All animals were measured (linear measurement after García-Barros;[36] Forewing length) and marked with an individual color code on the abdomen.

At least one hour before a male was released it was fed with 2 M sugar solution to assure that it had enough energy to fly. Experiments were only performed during warm summer nights without rain or strong wind. The amount of cloud cover was documented before the release of each male. During the course of the experiment, there were rarely many clouds (8.6% of flights) and never a completely overcast sky. We measured the time an individual needed to reach a trap with a stopwatch and caught each male directly afterwards to be stored safely until the next day when it was allowed to perform another flight. Thus, males that managed to arrive at a trap were allowed to perform further flights, but each male was tested only once per day. By applying this procedure, we were able to test 23 moths and 17 of the males performed more than one flight (see Fig. S1), resulting in a total of 58 departures from the release site. We determined the fraction of males arriving at a trap, which trap was selected and how long it took each arriving individual to get there. Except for red light used shortly before the release (to prepare the animals in some cases), no artificial light sources were used during the experiment. Whether red light was used or not depended on the specific experiment that differed in the handling procedure because the dataset analyzed here also served as the control experiment of another study. Since the handling procedure had no significant effect on the flight behavior of males, we pooled all flights for the analysis of the present study.

All-sky photometry was used to measure spatially resolved sky brightness and its natural and artificial component utilizing a commercial digital single-lens reflex camera (Canon EOS 6D) with a full-frame CMOS sensor (20.2 Megapixel) operating with a 180° circular fisheye lens (Sigma 8 mm f/3.5 EX DG). The camera was mounted on a tripod and positioned five meters away from the release site. Heating pads were attached to the camera to avoid the formation of dew on the

lens. Each night the camera was first aligned to the South and then tilted back to a 90-degree angle, so that the center of the lens was oriented towards the zenith. Images were obtained with ISO 3200 and varying shutter speed (15 s or 30 s) at intervals of 1 min. For the analysis, the first image obtained after the departure of each moth from the release site was processed. From the CR2 images (raw image format, Fig. 2a) luminance ($L_v$ unit mcd/m²) was calculated for each pixel with the software "Sky Quality Camera" (latest version 1.8.1, Euromix, Ljubljana, Slovenia, Fig. 2b, c).

From this data, $L_v$ was calculated for twelve defined sectors along the azimuth direction with 30° width (Fig. 2c, extended dashed lines). For days when the moon was almost full (95.7%), five extreme outliers (values outside the range of the median ± 4 times the median absolute deviation) were excluded from the analysis. This concerned all sectors except sector four. For analyses when the moon was below the horizon three extreme outliers were excluded (one in sector two and two in sector eleven). Additionally, the sky was segmented into further regions of interest, namely an outer ring of 5° elevation from the horizon (85°–90° zenith angle) and an inner ring 20° above the outer ring (65°–85° zenith angle; Fig. 2c)). This segmenting allowed to distinguish the brightening of the sky within the outer ring (dominated by ALAN) and the brightening within the inner ring (dominated by the moon) and how this affected flight behavior. The luminance is reported in "natural sky units" (NSU), which is more intuitive because a value in NSU indicates how much brighter or darker the sky was compared to a non-light polluted moonless clear night sky. It is defined here as 1 NSU ≈ 0.25 mcd/m² at zenith[37], please note that the night sky luminance is also slightly elevated near the horizon for non-light polluted sites[38].

The position of the moon defined by elevation and cardinal direction was retrieved from https://www.timeanddate.de/. We used the all-sky pictures to determine the horizontal profile. As the landscape in the east was quite flat, the moon was visible at an elevation of one degree and was considered to be above the horizon after passing this threshold.

**Statistics and reproducibility**. Analyses were conducted with the R statistical programming environment version 4.0.3[39]. We evaluated the arrival probability of moths within eight minutes after release ($n = 58$) in a logistic regression model (function glmer from R package lme4, version 1.1-26) with moon elevation, clouds in front of the moon, forewing length, as well as origin (breeding vs. field) as potential fixed-effect predictors, and individual moth as random factor. Flight duration of moths arriving at a trap within eight minutes after release ($n = 34$) was evaluated in a Cox Proportional Hazard survival model (function gam from R package mgcv, version 1.8-33, and link function cox.ph) with the same predictors and percentage of lit moon disk. In addition, we calculated median survival time (i.e. flight duration) from estimated survival function for arrivals within 8 minutes after release in response to moon elevation to visualize model predictions averaged over individuals. Choice of female trap (north or south) was modelled for all males that arrived at a trap ($n = 42$) by logistic regression in response to moon position (measured as the angle between the northern direction and the projection of the connection line of the moon and center of an experiment to the ground), clouds in front of the moon, breeding conditions as potential fixed-effect predictors, and individual moth as random factor. This analysis was repeated with the moon position replaced by the radial position of the section with maximum $L_v$ in elevation classes for the outer and the inner ring ($n = 41$).

**Ethical Note**. Our study involved individuals of *S. ligustri*, which were either reared by ourselves or trapped in the wild. We obtained permission for capture and release from the Regional Council of Giessen, Germany. All moths were carefully handled during experiments and maintained under appropriate conditions.

**Reporting summary**. Further information on research design is available in the Nature Research Reporting Summary linked to this article.

## Data availability

The dataset used in this study as well as the raw and processed all-sky pictures are available on DRYAD at https://doi.org/10.5061/dryad.wdbrv15qn[40].

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

## Acknowledgements

We thank Benjamin Lee and Thomas Walter for help in the field as well as the farmers in Großseelheim who gave us access to their grassland. Furthermore, we would like to thank Professor Menzel for the possibility to perform the nighttime experiments on the same experimental field his group used during the day. We are very grateful to Anna Stöckl for her comprehensive advice on establishing and maintaining the breeding of the moths. We also thank Basil el Jundi and Gerrit A. Thiene for their helpful comments on our manuscript. Funding was provided by DFG DE 2869/1-1. This publication was supported by the Open Access Publication Fund of the University of Würzburg.

## Author contributions

J.D. designed the study, M.S., A.Ja. and J.D. performed the experiment and extracted the data. M.S. and J.D. acquired and evaluated the data to quantify the light environment with substantial contributions of A.Je. and F.H. Data were analyzed by M.S., O.M., T.D., T.H. and J.D. The original drafting of the article was done by M.S. and J.D. and all authors contributed to the editing of subsequent drafts.

## Funding

So far written in the Acknowledgements: Funding was provided by DFG DE 2869/1-1. This publication was supported by the Open Access Publication Fund of the University of Würzburg. Open Access funding enabled and organized by Projekt DEAL.

## Competing interests

The authors declare no competing interests.
