## [Peer Review File · Communications Biology]

Reviewers' comments:

Reviewer #1 (Remarks to the Author):

OVERALL ASSESSMENT:

This is an interesting paper on a topic of both basic and applied biological significance that has been much neglected: the effect of the moon on orientation behavior of nocturnal insects, and how it might be affected by light pollution. While many studies at this point have examined the influence of light pollution on nocturnal insects, few have accounted for variation in the brightness of the moon, and the role that this natural source of night light plays. Thus, the premise of the study by Storms et al. is very timely and welcome. The experiment is a relatively simple one that appears to have been undertaken properly, the statistical analyses appear appropriate, and the authors have gone to great lengths to examine moon brightness and direction in detail. The results regarding moon position effects are obvious, interesting, and important. However, possibly due to the choice of location at which the study was conducted, there appears to have been no real effect of light pollution on moth orientation behavior, with the moon being the much more important source of light. This is an important point but is not well reflected in the Discussion, which presents too much conjecture regarding "possible effects" of light pollution, when none really appear to have been found. This should be amended. In addition, while I realize that space constraints have necessitated much detail to be placed in the Supplementary Material instead of the main text, some highly critical information, such as sample sizes, study duration, etc, are missing from the main text which makes it hard for a reader of the main text only to assess. These basic facts should be included in the main text in my opinion. Finally, Figure S2 was missing from the Supplemental Materials, so I was unable to assess it. I have included a list of specific comments below.

SPECIFIC COMMENTS:

Main Text

- Some more background and references are needed in the Introduction regarding what is known about the orientation of nocturnal insects, and moths in particular, and what we know currently about the cues nocturnal insects use. In other words, why should we even be looking at the position of the moon in this study? In addition, you appear to be missing a critical recent reference on this general topic: Foster et al. 2021. Light pollution forces a change in dung beetle Orientation behavior. Current Biology <https://www.sciencedirect.com/science/article/pii/S0960982221008332>
- I realize, due to space constraints in this particular journal, that a lot of details have to be left to the Supplementary Materials and Methods, but it would be useful to have a few more methodological details in the main text, especially things like sample size (right now I have no idea if the results in this paper are referring to 3 moths or 300!), the number of days the experiment was run, and a better quantification and description of local light pollution (where were the closest sources of ALAN?). Details found in Lines 5-16 of the Supplementary Methods should probably be included in the main text to allow readers to quickly judge the significance and context of your results.
- Line 61: "In addition" to what?
- Line 112: Fix grammar ("allow to track changes")
- Line 115-116: This appears to be somewhat conjectural, especially given the fact that moths were apparently not attracted to sources of light pollution according to your results (line 110). Could an alternative explanation be that there is high variance simply in how important moon position is to the orientation ability of individual males? Or perhaps this sentence would be equally true just by removing "light pollution" and simply stating "The high variance of flight durations in this constellation suggests that a lacking moon contributes to the fact that...". Could the authors please provide evidence that this alternative is not possible? Why is light pollution necessary here?
- Line 129-131: Again, this appears to be conjecture ("Light pollution near the horizon has potentially masked the moon near the horizon.."). Do you not have the data to test this definitively? It appears that your data do not necessarily support such an assertion.
- General Point: Did you compare the spectral properties of the moon compared to local light pollution? Insect vision is dependent on the spectral properties of light, not just overall illuminance.
- Lines 140 and 152: Do you mean "breeding colony"?

Supplementary Methods:

- Lines 5-16 should be included in the main text
- Lines 7-8: Were any sources of ALAN visible at the site? (streetlights, buildings, etc?). This statement about very low effects of ALAN on night sky brightness also undermines somewhat the conclusions offered at the end of the Discussion regarding ALAN possibly impacting orientation behavior by obscuring the moon at low horizons.
- Lines 13-14: describe what you mean by "under room conditions". This is very vague.
- Line 18-19: What do you mean by "good and stable"?
- Figure S2 does not appear to be included in the Supplementary Materials document, so I have no way of assessing it.

Reviewer #2 (Remarks to the Author):

The authors characterized nocturnal orientation of moths (reproductive context) at different moon elevations. This was measured by releasing males 105m from caged females and recording success rate & speed under different lunar positions. Males showed higher success rates at finding females when the moon was above the horizon and were quicker at success the higher the moon was above the horizon. The authors also analyzed the nighttime sky's light patterns by processing luminance levels of images of the sky during testing. Authors suggest the high variance in individual's delay to arrive at the females is due to the potential for light pollution present low in the sky to interfere with moon-based orientation when the moon was at low elevations.

The authors show that the moon aids in moth orientation, and postulate that the increased variance in success speed when the moon is near the horizon is due to the presence of light pollution. The authors theme these findings around the potential for light pollution at the horizon to interfere with orientation. An interesting finding to be sure, and as the authors conclude, one that would likely require further experimentation to state confidently.

Orientation success appears rather quick in most tests even when the moon is below the horizon in Fig. 1. Additionally, the authors state that cloud cover occluding the moon's position does not decrease success. Yet the authors make no statement as to what could be aiding orientation in these conditions despite stating the pheromone cannot facilitate quick orientation. It seems strange to have multiple instances where the moon is not visible, observe rapid orientation and then make no comment on the potential cues. What role, if any, could polarized lunar light play in orientation when the moon is blocked?

On the whole, the manuscript is well written; however, I have compiled a small number of minor issues for the authors to attend to.

Minor comments:

The Abstract/Summary seems very stilted and flows poorly. As this is not the case with the rest of the manuscript, my guess is that this may have arisen from cutting for word count. I have added some comments to improve the readability of this section.

Line 13: '...by the potential driver light pollution.'

Consider rewording

Line 14: 'We show that the utility of the moon as an orientation cue depends on its elevation, indicating that essential properties for orientation might have been obscured at lower moon elevations close to the horizon where light pollution was strongest.'

Repetitive. The sentence is structured strangely and does not flow well.

Line 20: "European countries" can be shortened to "Europe".

Line 25-27: While the impact of ALAN on nocturnal pollination has recently been described shown [10], its influence on the orientation of moths in the context of reproduction is so far poorly understood.

Sentence needs some work. I have made some suggestions.

Line 27: Here we present our findings on the effect of moonlight and surrounding light pollution of the surroundings on male mate finding success in male privet hawk moths (*Sphinx ligustri*).

This is overselling the current study's findings regarding light pollution.

Figure 1 shows four data points near -20deg but these are not visible in SFig 1. Am I missing something as to why the x-axis of these figures is different?

Figure 1's y-axis is the first time the word "trap(s)" has been mentioned. Confusing to readers.

Line 61: This is awkward phrasing, consider rewording.

Line 66: "enhancing..." word choice?

Line 86: "...the flight direction of males."

Figure 2. Why are the cardinal directions inside panel d?

Line 89: "...behavior of male moths."

Line 94: delete space between) and ,

Line 99: "Male moth affinity..."

Line 112: "...and allow researchers to track..." there is a missing word between "allow" and "to".

Line 116: 'Constellation' seems like a strange word choice and might confuse readers.

Line 124-125: This is awkward phrasing. Consider rewording.

Line 133: "protect" feels like the wrong word to use here but I can see how one could argue for this framing. I'll leave this to the authors' discretion.

Line 140: "...the breeding of the moths."

Reviewer #3 (Remarks to the Author):

In this article the authors test the role of lunar elevation on male moth mate finding ability at night. They also investigate the effects of low levels of artificial light at night. Their study design is simple but elegant - they released male moths and then measured the time it took for males to reach one of two "pheromone" traps, that each had 3 females in it. One trap was to the north and one was to the south. They did indeed find that lunar elevation predicted male mate finding success, with higher lunar elevation resulting in both more shorter duration flights to find mates and overall higher success. Furthermore, and much to their credit, they include measurements of night sky brightness using a CCD camera and they analyze the hemispherical night light conditions. This is another strength of their paper. Overall, the paper has many strengths and the story merits publication in a high impact journal like Nature Communications Biology. However, I do think there are several areas where the paper can be strengthened.

1) I do think that the paper could be expanded a little more. If I understand correctly, articles in Nature Comm Bio can be 5,000 words long, in which case the authors have much more space for

expanding on their study. If I am correct, then I suggest the authors attempt to highlight more the hypotheses involved as well as predictions. I don't understand why they did a trap to the north and to the south - what was the point behind this? What were you testing? Also, and more importantly, I think the biggest strength of this paper has nothing to do with artificial light but instead showing that moths do indeed use the moon for orientation - which has been a controversial topic in entomology and behavioral ecology for over a hundred years - THIS NEEDS TO BE BUILT UP MORE! I am afraid many people won't see this paper because it is focused more on light pollution - so the authors need to stress this. There is much history in the question of moth orientation relative to the moon and I think the authors need to highlight this.

2) More methods are needed in the main text - especially the overall numbers of male moths and how many times each moth was run. How repeatable were moths and if a moth didn't come to the trap, were they just lost for ever?

3) The figure legends could be much more helpful. I still don't really understand what is going on in f and g.

4) Another discussion point is needed and that is that you don't know if the moon is helping guide orientation due to acting as a landmark or just because the environment is 100 times brighter. Please discuss this and what research is known about this.

5) Again, in the methods, and preferably in the main text, I really would like to see more details on the lunar phase. I don't think I saw anywhere what the specific illumination of the moon was during the experiment. Also, why didn't you include percent illumination of the moon in your model? I assume the closer the moon was to full moon, the better the moths could find mates. I do encourage you to include this in your model or explain why this isn't necessary.

6) When you did repeated flights of the same male, were they on the same day or different day?

Some minor comments

13) "potential driver light pollution" doesn't make sense, reword please

68) Define NSU

For figure 2d) why not put the cardinal direction angle on the x axis and also place an icon of the moon on that scale like you did with the cardinal directions? That would make it very clear why that one bar is so bright.

152-153) I think you are missing a word after "breeding"

Overall, this is a really great study and I commend you for your efforts! I look forward to seeing a revised manuscript!

Response to the Reviewers:

To Reviewer #1:

1. OVERALL ASSESSMENT:

This is an interesting paper on a topic of both basic and applied biological significance that has been much neglected: the effect of the moon on orientation behavior of nocturnal insects, and how it might be affected by light pollution. While many studies at this point have examined the influence of light pollution on nocturnal insects, few have accounted for variation in the brightness of the moon, and the role that this natural source of night light plays. Thus, the premise of the study by Storms et al. is very timely and welcome. The experiment is a relatively simple one that appears to have been undertaken properly, the statistical analyses appear appropriate, and the authors have gone to great lengths to examine moon brightness and direction in detail. The results regarding moon position effects are obvious, interesting, and important. However, possibly due to the choice of location at which the study was conducted, there appears to have been no real effect of light pollution on moth orientation behavior, with the moon being the much more important source of light. This is an important point but is not well reflected in the Discussion, which presents too much conjecture regarding “possible effects” of light pollution, when none really appear to have been found. This should be amended. In addition, while I realize that space constraints have necessitated much detail to be placed in the Supplementary Material instead of the main text, some highly critical information, such as sample sizes, study duration, etc, are missing from the main text which makes it hard for a reader of the main text only to assess. These basic facts should be included in the main text in my opinion. Finally, Figure S2 was missing from the Supplemental Materials, so I was unable to assess it. I have included a list of specific comments below.

Response: We thank the reviewer for the nice words and for helping to find the right focus. We made several changes throughout the manuscript to shift the focus from ‘light pollution’ to ‘the importance of the moon’. Furthermore, we moved the methods section from the Supplementary Information to the main text and added the requested critical information. Also, the mistaken reference to the obsolete Fig. S2 has been deleted. Please see also our comments below.

SPECIFIC COMMENTS:

2. Main Text

- Some more background and references are needed in the Introduction regarding what is known about the orientation of nocturnal insects, and moths in particular, and what we know currently about the cues nocturnal insects use. In other words, why should we even be looking at the position of the moon in this study? In addition, you appear to be missing a critical recent reference on this general topic: Foster et al. 2021. Light pollution forces a change in dung beetle Orientation behavior. Current Biology
<https://www.sciencedirect.com/science/article/pii/S0960982221008332>

Response: Thank you for pointing this out. We reworked the introduction and now also mention the excellent study by Foster et al. (which had not yet been published at the time of initial submission) in the introduction as well as in the methods since it also uses all-sky

cameras to track nocturnal light while studying insect behavior. We believe that we now steer the reader much better towards our research questions.

3. I realize, due to space constraints in this particular journal, that a lot of details have to be left to the Supplementary Materials and Methods, but it would be useful to have a few more methodological details in the main text, especially things like sample size (right now I have no idea if the results in this paper are referring to 3 moths or 300!), the number of days the experiment was run, and a better quantification and description of local light pollution (where were the closest sources of ALAN?). Details found in Lines 5-16 of the Supplementary Methods should probably be included in the main text to allow readers to quickly judge the significance and context of your results.

Response: Thank you for these valuable suggestions. We have indeed kept the original version of this manuscript so short due to space constraints and are very pleased to extend the length of the main text according to your suggestions. Due to the transfer of the manuscript to Communications Biology we have much more space than before, which is why we moved the entire methods section from the Supplementary Information to the main text. Furthermore, we added detailed information about sample size throughout the manuscript and included a quantification description of the light sources in the surroundings.

4. Line 61: “In addition” to what?

Response: We meant in addition to natural light sources. However, we realized that the whole sentence was indeed not very clear and rephrased it. We now write: “This relatively high variance at low moon elevations emphasizes the question if artificial lights affected mate finding, particularly whenever the moon as a natural light cue was not yet prominent.” Page 5, lines 109-113

5. Line 112: Fix grammar (“allow to track changes”)

Response: We followed the suggestion of Reviewer #2 and fixed the grammar by inserting “researchers”. Page 10, line 200

6. Line 115-116: This appears to be somewhat conjectural, especially given the fact that moths were apparently not attracted to sources of light pollution according to your results (line 110). Could an alternative explanation be that there is high variance simply in how important moon position is to the orientation ability of individual males? Or perhaps this sentence would be equally true just by removing “light pollution” and simply stating “The high variance of flight durations in this constellation suggests that a lacking moon contributes to the fact that...”. Could the authors please provide evidence that this alternative is not possible? Why is light pollution necessary here?

Response: We agree that it is not necessary to discuss this finding explicitly in the context of light pollution. However, we think that this hypothesis is noteworthy and believe that our changes to eliminate the overstatement of light pollution throughout the manuscript prevent overemphasis on this aspect. Nonetheless, we realized that we did not explain properly what we mean and therefore rephrased this section.

We now write: “Digital cameras are suitable to measure the dynamics of night-time lighting conditions^{25,26}, and allow researchers to track changes in artificial lighting conditions and brightness of the sky simultaneously²⁷. However, it is not straightforward to distinguish between ALAN and natural light sources like the moon with luminance images when the moon is close to the horizon and thus in the section of the sky where most light pollution occurred. Yet, once the moon rose higher than 5° and thus stood out distinctly from the light-polluted horizon, it could be clearly identified on the images (Fig. 2b). In this context, it is particularly

remarkable that the speed at which the females were reached increased reliably only above a similar threshold (Fig. 1), with the only exceptions of two flights with long durations at a moon elevation greater than 20° (Fig. 1); both flights originated from the same individual (Fig. S1). Thus, the high variance of flight durations at low moon elevations (Fig. 1) supports our hypothesis that the moon, as an orientation cue, can be masked by artificial light for the animals as well. Yet, this hypothesis needs to be explicitly tested in future experiments.” Page 10, lines 199-212

7. Line 129-131: Again, this appears to be conjecture (“Light pollution near the horizon has potentially masked the moon near the horizon..”). Do you not have the data to test this definitively? It appears that your data do not necessarily support such an assertion.

Response: We agree. Our aim was to propose this explanation in particular because we consider it to be a conclusive hypothesis that can be specifically tested in future experiments, please see also our comment to point 6 above. Since we have elaborated this hypothesis in more detail in the discussion now, we deleted the respective sentence to prevent misunderstandings.

8. General Point: Did you compare the spectral properties of the moon compared to local light pollution? Insect vision is dependent on the spectral properties of light, not just overall illuminance.

Response: In this study we focused on the importance of the moon as an orientation cue and the question if artificial light sources of the surroundings can elicit a comparable effect on the behavior of moths. We did not compare the spectral properties of the moon compared to local light pollution.

9. Lines 140 and 152: Do you mean “breeding colony”?

Response: Yes, we mean we reared the moths by ourselves. To make this more clear, we added “of the moths” in the former line 140 (now page 16, line 353) and changed “which were either taken from our breeding”(former line 152) to “which were either reared by ourselves” (now page 16. Lines 365-366).

Supplementary Methods:

10. Lines 5-16 should be included in the main text

Response: Done, please see also our comment to point 1 above.

11. Lines 7-8: Were any sources of ALAN visible at the site? (streetlights, buildings, etc?). This statement about very low effects of ALAN on night sky brightness also undermines somewhat the conclusions offered at the end of the Discussion regarding ALAN possibly impacting orientation behavior by obscuring the moon at low horizons.

Response: We apologize for our non-ideal phrasing here. No direct and near light sources were visible from the site, just at the distance, and light pollution at zenith was not very strong. However, light pollution in form of skyglow from nearby settlements resulted in elevated sky brightness near the horizon as confirmed by our camera-based light measurements. We rephrased this section, please see also our comments to point 6 & 7 above.

We now write:” A skyglow model based on satellite data suggests that the night sky brightness at zenith was relatively low (0.3 mcd/m^2)¹⁹, which is indicated by the visibility of the Milky Way (Fig. 2c) and confirmed by our own measurements from such all-sky imagery for moonless clear nights ($0.34 \text{ mcd/m}^2 \approx 1.36 \text{ NSU}$). Skyglow is a type of indirect light

pollution that originates from upwards radiated light that is then scattered back within the atmosphere¹⁹. The surrounding villages and towns utilize standard artificial light sources (streetlights, private lights etc.). Therefore, some direct light sources as well as skyglow were visible at the (distant) horizon from our site, originating from the adjacent village Großseelheim (0.4km), the nearby cities Kirchhain (4.2km), Amöneburg (5.0km), Marburg (7.8km), Stadtallendorf (10.0km), and Giessen (30.0km). It is important to note that the setup was located far beyond the attraction radius, i.e. the distance around a lamp at which an animal is directly attracted towards the light source, of any artificial light in the surroundings, as these were much further away than 23m³⁵.” Pages 11-12, lines 246-259

12. Lines 13-14: describe what you mean by “under room conditions”. This is very vague.

Response: We agree that this description is very vague. However, we neither manipulated nor measured the temperature or humidity in the room where the pupae were kept for hatching since the moths hatched reliably. Nevertheless, it is certainly useful to provide some more information here as requested.

We therefore changed the respective sentences to: ”16 of the 23 males tested came from our own breeding, comprising the offspring of a mated female captured in the same study area one year before. These pupae raised were stored during winter in the refrigerator at a temperature of 5°C and removed five weeks before the experiment. For hatching, the pupae were placed on the bottom of a cardboard box, as this allowed them to climb the walls and unfold their wings easily. Spatial separation was used to ensure that males and females were unable to mate. The hatching boxes were kept in a room without thermoregulation, so that humidity and temperature fluctuated between day and night according to the warm summer temperatures.” Pages 12-13, lines 270-277

13. Line 18-19: What do you mean by “good and stable”?

Response: We replaced “good and stable weather conditions in the late evening and night” with “warm summer nights without rain or strong wind”. Page 13, lines 282-283

14. Figure S2 does not appear to be included in the Supplementary Materials document, so I have no way of assessing it.

Response: In an earlier version of the manuscript, some figures were organized differently. Unfortunately, we missed to amend the references to Fig. S2, a figure that no longer exists, when we restructured the manuscript. We have now deleted these references.

Reviewer #2 (Remarks to the Author):

15. The authors characterized nocturnal orientation of moths (reproductive context) at different moon elevations. This was measured by releasing males 105m from caged females and recording success rate & speed under different lunar positions. Males showed higher success rates at finding females when the moon was above the horizon and were quicker at success the higher the moon was above the horizon. The authors also analyzed the nighttime sky’s light patterns by processing luminance levels of images of the sky during testing. Authors suggest the high variance in individual’s delay to arrive at the females is due to the potential for light pollution present low in the sky to interfere with moon-based orientation when the moon was at low elevations.

The authors show that the moon aids in moth orientation, and postulate that the increased variance in success speed when the moon is near the horizon is due to the presence of light

pollution. The authors theme these findings around the potential for light pollution at the horizon to interfere with orientation. An interesting finding to be sure, and as the authors conclude, one that would likely require further experimentation to state confidently.

Orientation success appears rather quick in most tests even when the moon is below the horizon in Fig. 1. Additionally, the authors state that cloud cover occluding the moon's position does not decrease success. Yet the authors make no statement as to what could be aiding orientation in these conditions despite stating the pheromone cannot facilitate quick orientation. It seems strange to have multiple instances where the moon is not visible, observe rapid orientation and then make no comment on the potential cues. What role, if any, could polarized lunar light play in orientation when the moon is blocked?

Response: Lunar polarization is an interesting point, thanks for raising this! We added this aspect to the introduction and the discussion.

In the introduction we now write:

"Nocturnal animals have evolved under natural nocturnal light and can therefore utilize dim nocturnal light for orientation¹¹, which includes navigation by starlight¹²⁻¹⁴. For example, nocturnal insects such as dung beetles can use the Milky Way as orientation cue¹⁵ but also sense polarization patterns from moonlight^{16,17}." Page 2, lines 37-41

In the Results and Discussion-section we now write:

"Another interesting next research project would be to investigate the relevance of polarized light, as this could provide an explanation for the occasional fast flights at times of low lunar elevations (cf. Fig. 1). Further, it might explain why flight duration was not significantly affected by clouds in front of the moon since the polarization pattern extends over the whole sky and is therefore not shielded completely by scattered clouds²⁸. For dung beetles it has been already shown that they are capable of using the polarization signal for navigation^{16, 29, 30} and it has been proposed that moths might be capable of utilizing the same signal³¹. At the same time, it has already been demonstrated that urban skyglow can diminish the lunar polarization signal³², making a detailed investigation of the interplay between these two factors and the significance for moth orientation particularly exciting to understand underlying mechanisms." Pages 10-11, lines 218-227

On the whole, the manuscript is well written; however, I have compiled a small number of minor issues for the authors to attend to.

Minor comments:

16. The Abstract/Summary seems very stilted and flows poorly. As this is not the case with the rest of the manuscript, my guess is that this may have arisen from cutting for word count. I have added some comments to improve the readability of this section.

Response: Thank you very much for your comments, we rephrased the abstract accordingly. Furthermore, we were able to extend the abstract since Communications Biology allows for more words (the manuscript was transferred to Communications Biology).

17. Line 13: '...by the potential driver light pollution.'
Consider rewording

Response: We deleted this expression, please see also our comment to point 16 above.

18. Line 14: 'We show that the utility of the moon as an orientation cue depends on its elevation, indicating that essential properties for orientation might have been obscured at

lower moon elevations close to the horizon where light pollution was strongest.’ Repetitive. The sentence is structured strangely and does not flow well.

Response: We rephrased this section, please see also our comment to point 16 above.

19. Line 20: “European countries” can be shortened to “Europe”.

Response: Corrected as suggested. Page 2, line 33

20. Line 25-27: While the impact of ALAN on nocturnal pollination has recently been described shown [10], its influence on the orientation of moths in the context of reproduction is so far poorly understood.

Sentence needs some work. I have made some suggestions.

Response: We rephrased this part of the introduction and now write: “Recently, it was shown that dung beetle behavior is affected by ALAN²¹ and the impact of ALAN on important ecosystem services such as nocturnal pollination was documented²². However, the underlying mechanisms and cues of nocturnal orientation of moths are still poorly understood. In particular, the nocturnal orientation in the context of mate finding remains largely unknown but is of utter importance as the survival and mating success of moths might decrease by ALAN-mediated degradation of such orientation cues²³.” Pages 2-3, lines 44-50

21. Line 27: Here we present our findings on the effect of moonlight and surrounding light pollution of the surroundings on male mate finding success in male privet hawk moths (*Sphinx ligustri*).

This is overselling the current study's findings regarding light pollution.

Response: We have rephrased the entire paragraph according to the guidelines of Communications Biology and deleted this sentence as part of the process. We believe that the new paragraph no longer oversells ‘light pollution’ but rather emphasizes the importance of the moon for orientation in moths.

*We now write: “In this study, we combined behavioral experiments performed with free-flying male moths (*Sphinx ligustri*) with a detailed quantification of the nocturnal light environment using an all-sky camera. This allowed us to study natural light sources like the moon and the stars as well as skyglow - a type of indirect light pollution that originates from upwards radiated light that is then scattered back within the atmosphere¹⁹. We find that the visibility of the moon above the horizon improves the ability of male moths to find females and that they succeed faster as the moon rises. Although the moon increases the brightness of the entire environment, the cardinal position of the moon significantly influences the flight direction of males, as they choose to fly more frequently towards (caged) females located in the same hemisphere as the moon. Since bright areas at the horizon illuminated by distant light sources or skyglow do not trigger a comparable behavior, the moon as a natural light source apparently plays a key role in the orientation of male moths.” Page 3, lines 57-68*

22. Figure 1 shows four data points near -20deg but these are not visible in SFig 1. Am I missing something as to why the x-axis of these figures is different?

Response: Fig. S1 displays only the flight duration of males that arrived at a trap (within 8 min) more than once. The three flight durations at -19° displayed in Fig. 1 do not appear in Fig. S1 because there was no further flight of these 3 animals. Since the information of the two figures is not meant to be compared with each other, we have chosen the x-axis of Fig. S1 to display the additional information, namely the sequence of flights for each animal with more than one flight, as best as possible.

To describe this more clearly in the manuscript, we added the critical information to the 'Supplementary Results and Discussion' and the caption of Fig. S1.

Supplementary Results and Discussion:

"To exclude the possibility that males with multiple releases reached a trap progressively faster due to their increasing experience, we analysed the data of males with more than one flight on an individual basis (Fig. S1)." Page 4, lines 93-95

Caption of Fig. S1

"Please note that, different from Fig. 1, only flights of individuals with more than one flight are displayed in this figure." Page 4, lines 91-92

23. Figure 1's y-axis is the first time the word "trap(s)" has been mentioned. Confusing to readers.

Response: We agree. We inserted the sentence "Since we used these females specifically to exploit their attraction effect, we refer to them as 'traps' in the following." (page 4, lines 74-75) at the beginning of the results section and changed "females" to "trap" where it was appropriate.

24. Line 61: This is awkward phrasing, consider rewording.

Response: We rephrased this sentence and now write: "This relatively high variance at low moon elevations emphasizes the question if artificial lights affected mate finding, particularly whenever the moon as a natural light cue was not yet prominent." Page 5, lines 109-113

25. Line 66: "enhancing..." word choice?

Response: We rephrased the sentence and now write: "During observation times, the course of the moon mainly progressed through the eastern part of the sky, affecting particularly the luminance values in the corresponding sectors (Fig. 2d)." Page 6, lines 123-126

26. Line 86: "...the flight direction of males."

Response: Corrected as suggested. Page 7, line 150

27. Figure 2. Why are the cardinal directions inside panel d?

Response: We added the cardinal directions in panel d to make it easier to extract the course of the moon, which was mainly in the eastern part of the sky. Furthermore, we thought that it also facilitates the reference to sub-figures a-c. Of course, it is not absolutely necessary to leave the cardinal directions in sub-figure d, but we still find the information helpful for the reasons mentioned above and have therefore not changed the figure. To show the relation between Fig. 2a-c and Fig. 2d more clearly, we added a north arrow to Fig. 2c. Please see the inserted Fig. 2 below.

28. Line 89: "...behavior of male moths."

Response: Corrected. Page 8, line 157

29. Line 94: delete space between) and ,

Response: Done. Page 9, line 163

30. Line 99: "Male moth affinity..."

Response: Corrected as suggested. Page 9, line 170

31. Line 112: "...and allow researchers to track..." there is a missing word between "allow" and "to".

Response: Corrected as suggested. Page 10, line 200

32. Line 116: 'Constellation' seems like a strange word choice and might confuse readers.

Response: We rephrased this sentence and deleted the word 'constellation'. We now write: "Thus, the high variance of flight durations at low moon elevations (Fig. 1) supports our hypothesis that the moon, as an orientation cue, can be masked by artificial light for the animals as well. Yet, this hypothesis needs to be explicitly tested in future experiments." Page 10, lines 209-212

33. Line 124-125: This is awkward phrasing. Consider rewording.

Response: We rephrased the sentence and now write: "Since flight duration decreased as a function of lunar elevation, we conclude that the moon contributes to mating success, especially when it can be easily perceived." Page 11, lines 230-232

34. Line 133: "protect" feels like the wrong word to use here but I can see how one could argue for this framing. I'll leave this to the authors' discretion.

Response: This sentence was deleted in the course of the revision of the manuscript.

35. Line 140: "...the breeding of the moths."

Response: Corrected as suggested. Page 16, line 353

Reviewer #3 (Remarks to the Author):

In this article the authors test the role of lunar elevation on male moth mate finding ability at night. They also investigate the effects of low levels of artificial light at night. Their study design is simple but elegant - they released male moths and then measured the time it took for males to reach one of two "pheromone" traps, that each had 3 females in it. One trap was to the north and one was to the south. They did indeed find that lunar elevation predicted male mate finding success, with higher lunar elevation resulting in both more shorter duration flights to find mates and overall higher success. Furthermore, and much to their credit, they include measurements of night sky brightness using a CCD camera and they analyze the hemispherical night light conditions. This is another strength of their paper. Overall, the paper has many strengths and the story merits publication in a high impact journal like Nature Communications Biology. However, I do think there are several areas where the paper can be strengthened.

36. 1) I do think that the paper could be expanded a little more. If I understand correctly, articles in Nature Comm Bio can be 5,000 words long, in which case the authors have much more space for expanding on their study. If I am correct, then I suggest the authors attempt to highlight more the hypotheses involved as well as predictions.

Response: We are very grateful for the positive feedback on our paper and the hint that articles in Nature Comm Bio can be 5,000 words long. Due to the transfer of the manuscript to Communications Biology, we have indeed much more space and are very happy about the recommendation to expand the paper a little more. We moved the Materials and Methods-section from the Supplementary Information to the main text and added information to further clarify things. We also made changes throughout the manuscript to better highlight the hypotheses and predictions involved.

37. I don't understand why they did a trap to the north and to the south - what was the point behind this? What were you testing?

Response: Since the moon was almost exclusively visible in the southern hemisphere, we had the chance to establish a choice experiment by positioning one trap in the north and the other one in the south. We expected a preference for the southern trap if the flight behavior of males got influenced by the position of the moon. If not, changes in flight behavior would need to be considered as independent of the moon's position and would then rather have to be attributed to other changes, e.g. the increase in overall brightness. To address this point more clearly, we added information to the Materials and Methods section as well as the Results and Discussion section.

In the Results and Discussion-section we now write:

“Due to the design of the experiment with one trap located in the north and the other in the south of a central release site, we were able to investigate the choice behavior of males, especially in respect of the possible influence of cardinal position of the moon as it was almost exclusively visible in the southern hemisphere of the sky (Fig. 2d). Although the moon continued to move south during the night, the moon's cardinal position never overlapped with the exact direction of the southern trap. The only parameter that had a significant effect on choice behavior was indeed the cardinal position of the moon (Fig. 2e, logistic regression, $z=-2.3$, $p=0.022$, $n=42$). The more southern the moon's position was, the more likely males flew to the southern trap. However, while some clouds in front of the moon had no significant effect on choice behavior ($z=0$, $p=1$, $n=42$), moon above the horizon showed a tendency to affect males ($z=-1.82$, $p=0.069$, $n=42$). The results indicate that despite the general increase of ambient brightness by the moon, it is its position that mainly influenced the flight direction of males. Thus, moths preferred a flight direction with the prominent compass cue ahead to steer their flight towards the females but it is important to emphasize that moon and trap had an angular difference of at least 23° (80.8° to the moon's mean cardinal direction). Therefore, males that chose to fly towards the southern trap did not fly directly towards the direction of the moon.” Pages 6-7, lines 132-154

In the Materials and Methods-section we now write:

“To create a choice scenario, one trap was located in the north, the other one in the south of the release site. By selecting the north-south-axis we took advantage of the circumstance that whenever the moon was above the horizon, it was almost exclusively visible in the southern hemisphere of the sky (85% of flights). The choice of the trap therefore also reflected whether males had chosen the hemisphere with the moon and thus indicated whether the cardinal position of the moon influenced flight behavior.” Page 12, lines 264-269

38. Also, and more importantly, I think the biggest strength of this paper has nothing to do with artificial light but instead showing that moths do indeed use the moon for orientation - which has been a controversial topic in entomology and behavioral ecology for over a hundred years - THIS NEEDS TO BE BUILT UP MORE! I am afraid many people won't see this paper because it is focused more on light pollution - so the authors need to stress this. There is much history in the question of moth orientation relative to the moon and I think the authors need to highlight this.

Response: We are very grateful for this advice and fully agree with it. We have therefore not only made changes throughout the manuscript but also changed the title to “The rising moon promotes mate finding in moths”.

39. 2) More methods are needed in the main text - especially the overall numbers of male moths and how many times each moth was run. How repeatable were moths and if a moth didn't come to the trap, were they just lost for ever?

Response: We moved the Materials and Methods-section from the Supplementary Information to the main text and added or highlighted the requested information.

In the Results and Discussion-section we now write:

“We found that the percentage of males arriving within the experimental time (eight minutes from release, 58.6% of flights) at a trap increased significantly with the appearance of the moon (logistic regression: $z=-2.06$, $p=0.04$, $n=58$) and did not depend on the presence of clouds in front of the moon ($z=-0.83$, $p=0.406$, $n=58$). A few males reached the females later during the experimental night (13.8% of flights) and were released again on the next day. Some males never reached a trap and could therefore not be tested again in the next days (27.6% of flights).” Page 4, lines 79-85

In the Materials and Methods-section we now write:

“We measured the time an individual needed to reach a trap with a stopwatch and caught each male directly afterwards to be stored safely until the next day when it was allowed to perform another flight. Thus, males that managed to arrive at a trap were allowed to perform further flights, but each male was tested only once per day. By applying this procedure, we were able to test 23 moths and 17 of the males performed more than one flight (see Fig. S1), resulting in a total of 58 departures from the release site.” Page 13, lines 286-291

40. 3) The figure legends could be much more helpful. I still don't really understand what is going on in f and g.

Response: We have revised figure 2f and 2g as well as the corresponding part of the legend and hope they are better understandable now. In detail, we changed the name of the x-axis in f and g and rephrased the respective figure legends. Furthermore, we moved the corresponding parts from the ‘Supplementary results and discussion’ to the main manuscript.

Fig. 2 Quantification of the light environment with all-sky imagery and its impact on flight behavior of moths. (a) Raw RGB all-sky image with clear sky and a visible moon 26° above the horizon at 119° azimuth angle, South-east (24 July 2019, 03:23). (b) Same image as in (a) with processed luminance values. (c) Processed all-sky image in luminance with clear sky, a visible milky way (green patches in a ‘ribbon-shape’ across the (blue) night sky), skyglow near the horizon, and a non-visible moon 0° above the horizon at 87° azimuth angle, East (24 July 2019, 0:25). The colors of the processed image correspond to the legend in (b). The black lines mark the sky segments used to quantify the light environment. The outer ring covers 5° above the horizon (85° - 90° zenith angle), the inner ring 20° above the outer ring (65° - 85° zenith angle). Furthermore, the sky was divided into 12 sectors of 30° width along the azimuth direction (extension by dashed line), starting with the sector marked with the small circle (counting clockwise). (d) Luminance in natural sky units (NSU) for each full sector of 30° . The moon icons indicate sectors in which the moon was visible, regardless of its phase. The size of each symbol encodes the rank of the frequency ($n=33$). (e) Trap choice of arrived males depending on the position of the moon at the moment of release on the north-south axis (north = 0°). The y-axis displays choice of the southern trap at 0.0 and of the northern trap at 1.0. $p=0.022$, $n=42$. (f) Male moth affinity to northern trap in response to the direction of maximum luminance measured in the outer ring of 5° . The line represents the prediction of the logistic model, providing a probability value for arriving at the northern trap (north prone = 1; south prone = 0). Each circle indicates an observed arrival, $p=0.753$, $n=41$. (g) Male moth affinity to northern trap as in (f) but with luminance measured in the inner ring of 20° , $p=0.065$, $n=41$.

41. 4) Another discussion point is needed and that is that you don't know if the moon is helping guide orientation due to acting as a landmark or just because the environment is 100 times brighter. Please discuss this and what research is known about this.

Response: We are sorry that we were not able to incorporate this issue into the discussion, as it was not clear to us, based on our understanding of the word 'landmark', what the discussion point should be. Our understanding is that the moon is a celestial cue that can be used as directional information in the context of compass orientation (Grob, R., el Jundi, B. & Fleischmann, P.N. Towards a common terminology for arthropod spatial orientation. Ethology Ecology & Evolution 2021). In addition, many animals use landmark cues for orientation, but a compass cue cannot be a landmark cue at the same time, because they are at different levels, so to speak.

To emphasize the role of the moon for orientation in general, we have added to the introduction: "The most easily perceived celestial body during the night is the moon, which, however, can be more difficult to use as a compass than the sun because of its variable, temporally limited visibility¹². Nevertheless, it can potentially be used by nocturnal insects for orientation¹¹." Page 3, lines 51-54

Furthermore, we realized that the critical part of the results and discussion section addressing the impact of the general increase in ambient brightness and the position of the moon was not phrased clearly.

We therefore restructured and rephrased this paragraph and now write: " Due to the design of the experiment with one trap located in the north and the other in the south of a central release site, we were able to investigate the choice behavior of males, especially in respect of the possible influence of cardinal position of the moon as it was almost exclusively visible in the southern hemisphere of the sky (Fig. 2d). Although the moon continued to move south during the night, the moon's cardinal position never overlapped with the exact direction of the southern trap. The only parameter that had a significant effect on choice behavior was indeed the cardinal position of the moon (Fig. 2e, logistic regression, $z=-2.3$, $p=0.022$, $n=42$). The more southern the moon's position was, the more likely males flew to the southern trap. However, while some clouds in front of the moon had no significant effect on choice behavior ($z=0$, $p=1$, $n=42$), moon above the horizon showed a tendency to affect males ($z=-1.82$, $p=0.069$, $n=42$). The results indicate that despite the general increase of ambient brightness by the moon, it is its position that mainly influenced the flight direction of males. Thus, moths preferred a flight direction with the prominent compass cue ahead to steer their flight towards the females but it is important to emphasize that moon and trap had an angular difference of at least 23° (80.8° to the moon's mean cardinal direction). Therefore, males that chose to fly towards the southern trap did not fly directly towards the direction of the moon." Pages 6-7, lines 132-154

42. 5) Again, in the methods, and preferably in the main text, I really would like to see more details on the lunar phase. I don't think I saw anywhere what the specific illumination of the moon was during the experiment. Also, why didn't you include percent illumination of the moon in your model? I assume the closer the moon was to full moon, the better the moths could find mates. I do encourage you to include this in your model or explain why this isn't necessary.

Response: We included more details on the lunar phase in the main text and included percent illumination of the moon in our model.

In the Results and Discussion-section we now write:

“Although the lunar phase changed during the period of the experiment from full moon to new moon, flight duration was not significantly affected by the percentage of the lit moon disk ($z=0.44$, $p=0.66$, $n=34$). Thus, the properties of the moon that affected flight duration of males were independent of the lunar phase.” Page 4, lines 91-95

In the Materials and Methods-section we now write:

“The experiment was performed during the lunar phases from full moon (19 July, 95.7% lit moon disk) to new moon (31 July, 0.2% lit moon disk).” Page 11, lines 245-247

43. 6) When you did repeated flights of the same male, were they on the same day or different day?

Response: Repeated flights of the same male were on different days. To make this more clear, we added the sentence “Thus, males that managed to arrive at a trap were allowed to perform further flights, but each male was tested only once per day” (page 13, lines 288-289) at the respective part of the methods, which are now completely included in the main text.

Some minor comments

44. 13) "potential driver light pollution" doesn't make sense, reword please

Response: We have revised the abstract, and deleted this phrase.

45. 68) Define NSU

Response: We moved the Materials and Methods section including the definition of NSU (page 14-15, lines 320-325) from the Supplementary Information to the main text.

46. For figure 2d) why not put the cardinal direction angle on the x axis and also place an icon of the moon on that scale like you did with the cardinal directions? That would make it very clear why that one bar is so bright.

Response: We changed the figure according to your suggestions. In detail, we replaced the number of the sectors with the angle of the central cardinal direction of each sector. Further, we inserted a moon icon in the sectors where the moon was present and chose the size of the icon according to the frequencies with the biggest icon for the sector with the highest frequency and the smallest icon for the sector with the lowest frequency. Please see the inserted Fig. 2 above.

47. 152-153) I think you are missing a word after "breeding"

*Response: We rephrased this sentence and now write: „Our study involved individuals of *S. ligustri*, which were either reared by ourselves or trapped in the wild.” Page 16. Lines 365-366*

Overall, this is a really great study and I commend you for your efforts! I look forward to seeing a revised manuscript!

REVIEWERS' COMMENTS:

Reviewer #1 (Remarks to the Author):

The authors have done an excellent job of thoroughly revising their manuscript based on my review. The present paper makes an important contribution to our understanding of the navigation system of nocturnal animals, and is better structured now to focus on this important contribution. There remains several grammatical issues throughout the manuscript, especially in the Introduction, which could use some English language editing. Otherwise this is an excellent paper which the authors should be commended on.

Reviewer #2 (Remarks to the Author):

I am happy with the revisions and responses given by the authors to my (and the other reviewer) comments.

Communications Biology - COMMSBIO-21-2188A

Response to the Reviewers:

To Reviewer #1:

The authors have done an excellent job of thoroughly revising their manuscript based on my review. The present paper makes an important contribution to our understanding of the navigation system of nocturnal animals, and is better structured now to focus on this important contribution. There remains several grammatical issues throughout the manuscript, especially in the Introduction, which could use some English language editing. Otherwise this is an excellent paper which the authors should be commended on.

Response: We thank the reviewer for the nice words. We have carefully checked and edited the grammar throughout the manuscript, especially in the introduction.

Reviewer #2 (Remarks to the Author):

I am happy with the revisions and responses given by the authors to my (and the other reviewer) comments.